# Protective Effect of a Fucose-Rich Fucoidan Isolated from *Saccharina japonica* against Ultraviolet B-Induced Photodamage In Vitro in Human Keratinocytes and In Vivo in Zebrafish

**DOI:** 10.3390/md18060316

**Published:** 2020-06-15

**Authors:** Wanchun Su, Lei Wang, Xiaoting Fu, Liying Ni, Delin Duan, Jiachao Xu, Xin Gao

**Affiliations:** 1College of Food Science & Engineering, Ocean University of China, 5th Yushan Road, Qingdao 266003, China; suwanccchun@163.com (W.S.); niliying12@163.com (L.N.); xujia@ouc.edu.cn (J.X.); xingao@ouc.edu.cn (X.G.); 2Department of Marine Life Sciences, Jeju National University, Jeju Self-Governing Province 63243, Korea; comeonleiwang@163.com; 3Marine Science Institute, Jeju National University, Jeju Self-Governing Province 63333, Korea; 4State Key Lab of Seaweed Bioactive Substances, 1th Daxueyuan Road, Qingdao 266400, China; dlduan@qdio.ac.cn; 5Key Laboratory of Experimental Marine Biology, Institute of Oceanology, Chinese Academy of Sciences, No. 7 Nanhai Road, Qingdao 266071, China

**Keywords:** ROS, Phaeophyta, carbohydrate, UVB irradiation, HaCaT cells, zebrafish

## Abstract

A fucose-rich fucoidan was purified from brown seaweed *Saccharina japonica*, of which the UVB protective effect was investigated in vitro in keratinocytes of HaCaT cells and in vivo in zebrafish. The intracellular reactive oxygen species levels and the viability of UVB-irradiated HaCaT cells were determined. The results indicate that the purified fucoidan significantly reduced the intracellular reactive oxygen species levels and improved the viability of UVB-irradiated HaCaT cells. Furthermore, the purified fucoidan remarkably decreased the apoptosis by regulating the expressions of Bax/Bcl-xL and cleaved caspase-3 in UVB-irradiated HaCaT cells in a dose-dependent manner. In addition, the in vivo UV protective effect of the purified fucoidan was investigated using a zebrafish model. It significantly reduced the intracellular reactive oxygen species level, the cell death, the NO production, and the lipid peroxidation in UVB-irradiated zebrafish in a dose-dependent manner. These results suggest that purified fucoidan has a great potential to be developed as a natural anti-UVB agent applied in the cosmetic industry.

## 1. Introduction

Human skin is one of the most fundamental organs which is directly exposed to the external environment. It acts as a barrier to prevent the body from external damage, including ultraviolet (UV) irradiation exposure [1]. Although UV is beneficial to the human body to a certain extent, the excessive exposure to UV will damage the ability of basal keratinocytes, which are responsible for maintaining skin homeostasis to resist UV-induced damage, which can lead to different skin diseases such as accelerated degradation of collagen, inflammatory reaction, epidermal hyperplasia and skin cancers [2,3]. Solar UV irradiation contains wavelengths from approximately 100 to 400 nm, but only UVB (280–315 nm) and UVA (315–400 nm) reach the terrestrial surface [4]. It is now commonly known that the exposure to solar UVB is the major factor causing keratinocyte damage, causing DNA mutations, which can induce photo-aging and skin cancer [5].

The UVB damages skin through stimulating the production of reactive oxygen species (ROS) [6,7]. The expression of apoptosis-related cytokines induced by excessive ROS can lead to skin oxidative damage, and cause apoptosis [8]. Therefore, inhibiting the excessive ROS generation and eliminating potential cancer-causing cells are important strategies that constitute photoprotection and inhibit apoptosis [9].

Currently, researchers are focusing on developing natural compounds to explore their protective effects on UVB-induced photodamage. Seaweed is a plant abundantly growing in the sea and is broadly distributed all over the world. It can endure strong sunlight without any structural damage, which indicates that seaweed can protect itself from photodynamic damage, including UVB irradiation [10]. Thus, many studies have been conducted to find bioactive extracts or active ingredients derived from seaweed, such as polyphenol, polysaccharide, fucoxanthin, by which the generation of UVB-induced ROS can be modulated [11,12,13].

*Saccharina japonica* (*S. japonica*), a brown seaweed, used to be known as *Laminaria japonica* (*L. japonica*), which is plentifully produced in Asian countries and used as a traditional Chinese medicine and functional food [14,15]. In addition, it contains plenty of biological compounds, including fucoidan. Previous studies have reported that fucoidan isolated from *S. japonica* has a variety of bioactivities, such as neuroprotective, anti-inflammation, antioxidant, antiviral, immunomodulatory, atherosclerosis mitigation and anticoagulant activities [16,17,18,19,20,21]. Some researchers have investigated the anti-UVB ability of fucoidan from brown seaweed, such as *Costaria costata*, *Fucus evanescens* and *Undaria pinnatifida* [22,23,24]; however, this is the first study on the anti-UVB activities of fucoidan extracted from *S. japonica*.

In our latest work, the structure and anti-inflammatory activities of a fucoidan fraction of LJSF4 from *S. japonica* were studied [25]. In this study, we further investigated its photo-protective activity. The possible mechanisms against UVB-induced photodamage were examined in vitro using human keratinocyte (HaCaT) cells and in vivo in a zebrafish model in order to provide evidence for its application as an anti-UVB agent in cosmetics.

## 2. Results and Discussion

### 2.1. Purification and Monosaccharide Determination of Fucoidan Fractions

Fucoidan fractions of LJSF1-LJSF4 were purified from *S. japonica* (Figure 1A), among which LJSF4 has the highest sulfate content of 30.72%. As shown in Figure 1B, its monosaccharide compositions were determined to be 79.49% of fucose and 16.76% of galactose, thus it is a fucose-rich fucoidan. In our recently published work, the structures of LJSF4 were analyzed by different spectroscopic methods and its excellent anti-inflammatory activities were studied in vitro and in vivo [25]. In order to achieve the application of this fucoidan, knowledge of its various bioactivities is required. Thus, its anti-UVB activity and mechanism both in vitro and in vivo were investigated in this study.

### 2.2. Effect of LJSF4 on UVB-Irradiated HaCaT Cells

#### 2.2.1. Effect of LJSF4 on Intracellular ROS Generation and Cell Death in UVB-Irradiated HaCaT Cells

Prior to evaluate the anti-UVB effect of LJSF4 on HaCaT cells, we detected its potential toxicity on HaCaT cells through the MTT viability assay. As shown in Figure 2A, the viability of HaCaT cells without LJSF4 treatment was 100%, and those treated with LJSF4 were above 90%. In addition, compared with the control group, cells treated with 50 μg/mL of LJSF4 even had higher cell viability. These results indicate that LJSF4 had no cytotoxicity on HaCaT cells and even exhibited positive effect on the cells at the concentration of 50 μg/mL.

Exposure of cells to UVB irradiation results in ROS excessive production and cellular damage, which is the cause of skin cancer and photoaging [26]. Therefore, suppressing the excessive production of ROS can protect the skin from the damage of UVB [27]. In this study, the HaCaT cell damage induced by UVB irradiation was examined by measuring the intracellular ROS generation and cell death. As shown in Figure 2B, the ROS level of UVB-irradiated cells was significantly increased compared to those of non-irradiated cells, while the ROS levels of those treated with different concentrations of LJSF4 were significantly decreased in a dose-dependent manner. In addition, the viability of cells irradiated by UVB was significantly decreased, and dose-dependently increased in LJSF4 treated cells (Figure 2C). These results indicate that LJSF4 possessed a protective effect against UVB-induced HaCaT cell damage. The anti-UVB activities of fucoidan extracted from other brown seaweed of *Costaria costata*, *Fucus evanescens* and *Undaria pinnatifida* have been investigated [22,23,24]. However, these seaweed resources are not as rich as *S. japonica,* which is the most abundant economic seaweed around the world [28]. In addition, fucoidan extracted from *C. costata* increase the cell viability by 1.77% and 4.94% at concentrations of 0.01 and 0.1 μg/mL, respectively, while the fucoidan was mildly cytotoxic at the concentration of 1 μg/mL [24]. However, LJSF4 was able to increase the cell viability by 16.14% at the concentration of 100 μg/mL with no cytotoxic effect. Therefore, LJSF4 possessed an excellent anti-UVB activity on HaCaT cells.

#### 2.2.2. Effect of LJSF4 on Apoptosis in UVB-Irradiated HaCaT Cells

Apoptosis is a strong response of cells to UVB-induced damage. Cells eventually undergo apoptosis to eliminate severely damaged cells. However, even if serious damage occurs, some cells can still escape apoptosis, then turn into cancer cells. Studies have found that inhibiting apoptosis plays an important part in the formation of cancer [29].

In order to measure the protective effect of LJSF4 from UVB irradiation, HaCaT cells were stained with Hoechst 33342, which is a cell-permeable DNA dye. Later, the nuclear morphology of cells was observed by fluorescence microscopy. The cell images are shown in Figure 3. We found that the amount of apoptotic body of cells with UVB irradiation was almost 2.5 fold greater than that of cells without UVB irradiation, while those of cells pre-treated with 25 to 100 μg/mL of LJSF4 were decreased in a dose-dependent manner significantly. Masaki et al. reported that UVB-induced apoptosis was related to the increase in intracellular ROS generation. Inhibiting the production of ROS contributed to improving the intracellular defense against oxidative stress, thereby reducing cell apoptosis [27]. Our study evaluated the levels of ROS and apoptosis with or without LJSF4 treatment, and intervened with different concentrations of LJSF4. This result indicates that LJSF4 possessed excellent protective effect against UVB-induced HaCaT cell apoptosis.

#### 2.2.3. Effect of LJSF4 on Bax/Bcl-xL and Cleaved Caspase-3 Levels in UVB-Irradiated HaCaT Cells

Apoptosis is a regulated mechanism of cell suicide, usually manifested as nuclear condensation, wrinkling, membrane foaming as well as chromosomal DNA fragmentation [29,30,31]. According to the previous studies, the formation of apoptosome is an intrinsic apoptotic signaling pathway which could activated by oxidative stress [32,33,34]. Excessive ROS production as a second messenger regulates apoptotic signaling pathway following UV irradiation. In this case, the balance of anti-apoptotic molecules of Bcl-xL and pro-apoptotic molecules of Bax indicated whether apoptosis is promoted or suppressed [35]. Eventually, their imbalance triggered the caspase cascade [36]. In addition, cleaved caspase-3 directly induces cell death as a critical executor of apoptosis [37].

The Bax/Bcl-xL and cleaved caspase-3 levels of UVB-irradiated HaCaT cells were measured by Western blot analysis. As shown in Figure 4, the UVB irradiation increased the Bax level and decreased the Bcl-xL level, while pre-treatment with LJSF4 significantly increased the Bcl-xL level and decreased the Bax level in a dose-dependent manner. Moreover, LJSF4 reduced UVB-induced cleaved caspase-3 activation in UVB-irradiated HaCaT cells. These results indicate that LJSF4 inhibited UVB-induced apoptosis by the regulation of Bax, Bcl-xL and cleaved caspase-3 levels. The anti-UVB activity of Sargachromenol extracted from the brown seaweed of *Sargassum micracanthum* has been investigated, and the similar pathway by regulation of Bax, Bcl-xL and cleaved caspase-3 levels has been reported [38]. In addition, fucoidans isolated from different brown algae have been reported to inhibit UVB-induced photoaging through other signaling pathways, such as by inhibiting the MAPK pathways related to NF-κB and AP-1, by inhibiting MMP-1 expression via blocking the signal pathways of p38, JNK, and ERK [23,39]. Thus, the *S. japonica* fucoidan showed a different anti-UVB pathway to those of other reported fucoidan via regulating the expression of Bax, Bcl-xL and cleaved caspase-3.

### 2.3. Effect of LJSF4 on UVB-Irradiated Zebrafish

Zebrafish has become a popular animal model in the field of biological activity, due to the fact that its tissues and organs are very similar to mammals at the genetic, physiological, behavioral, and anatomical levels [40]. In previous reports, researchers have successfully used the zebrafish model to explore the protective effects of natural compounds on UVB-induced oxidative stress [41]. Therefore, we chose the same model to evaluate the protective effect of LJSF4 on UVB-irradiated photoaging in vivo. Zebrafish were pre-treated with LJSF4 of 25 to 100 μg/mL, respectively, and subsequently exposed to 50 mJ/cm^2^ of UVB.

Excessive ROS produced by UVB stimulation plays a critical role in destroying keratinocytes through cell damage. ROS can be detected by using 2′, 7′-dichlorodihydrofluorescein diacetate (DCFH-DA) staining in living embryos. As shown in Figure 5A, the control group without exposure to UVB had no fluorescent generation, whereas that with exposure to UVB generated fluorescence, which indicated the generation of ROS on UVB-irradiated zebrafish. However, the zebrafish treated with different concentrations of LJSF4 showed a dose-dependent reduction in the production of ROS before being exposed to UVB irradiation. This result indicates that ROS levels decreased by 237.64% in zebrafish treated with 100 μg/mL of LJSF4. Then, cell death was determined by staining the embryos with acridine orange. The result indicates that the fluorescence intensity of the LJSF4 pretreatment group at the concentration of 100 μg/mL was decreased to almost the same level to that of the control group (Figure 5B). In addition, the protective effect of the fucoidan on zebrafish was determined against ROS-activated NO generation. As shown in Figure 5C, compared to that of the control group with no UVB irradiation, the production of NO was significantly increased 3.64 fold, while the pretreatment with LJSF4 inhibited the production of NO in a dose-dependent manner significantly. A similar result of dose-dependent inhibition for lipid peroxidation by LJSF4 treatment was also obtained (Figure 5D). This is similar to the results of previous studies, which reported that zebrafish exposed to UVB increased ROS generation, cell death levels, NO generation, and lipid peroxidation levels compared to that without UVB irradiation [12,42]. Furthermore, Wang et al. (2017) demonstrated that DPHC, isolated from *I. okamurae*, could decrease ROS levels by 85.21% with the concentration of 100 μM [41]. In short, the results show that LJSF4 extracted from *S. japonica* could excellently inhibit the generation of inflammation and reduce the destruction of cellular components by decreasing ROS levels, thereby further indicating the photoprotective effect of LJSF4 in zebrafish.

## 3. Materials and Methods

### 3.1. Reagents and Chemicals

*S. japonica* was harvested in Xiapu coastal area, Fujian province, China. Phosphate-buffered saline (PBS) and fetal bovine serum (FBS) were purchased from Solarbio (Beijing, China). The dimethyl sulfoxide (DMSO), DCFH-DA, 1,3-Bis (diphenylphosphino) propane (DPPP), dimethyl sulfoxide (DMSO), diaminofluorophore 4-amino-5-methylamino-2′, 7′-difluorofluorescein diacetate (DAF-FM-DA), bovine serum albumin (BSA), 3-(4, 5)-dimethylthiazol-2-yl)-2, 5-diphenyltetrazolium bromide (MTT), acridine orange, and Hoechst 33342 were purchased from Sigma Co. (St. Louis, MO, USA). Penicillin/streptomycin (P/S) and Dulbecco’s modified Eagle’s medium (DMEM) were purchased from Gibco (Rockville, MD, USA). Antibodies against GAPDH (clone number of ARC0205, catalog number of MA5-35235), Bax (clone number of ARC0164, catalog number of MA5-35342), Bcl-xL (clone number of C.85.1, catalog number of MA5-15142), and cleaved caspase-3 (clone number of ARC0133, catalog number of MA5-35333) were purchased from Thermo Scientific (Waltham, MA, USA). All other reagents used in this study were of analytical grade and purchased from Solarbio (Beijing, China).

### 3.2. Purification and Monosaccharide Determination of Fucoidan Fractions

#### 3.2.1. Extraction and Purification of Polysaccharides

LJSF4 was prepared according to the method described in our previous study [25]. In brief, the defatted sample of *S. japonica* was extracted by triple volume of distilled water at 120 °C for 2 h, and then the supernatant was collected. Then, via adding an equal volume of 2% CaCl_2_ solution in order to remove the alginate in the supernatant, and the resulting supernatant was finally lyophilized to gain crude fucoidan of LJS. The crude LJS was further purified by an anion exchange chromatography (AKTA Purifier UPC100, GE healthcare, Pittsburgh, PA, USA) to obtain the four fractions of LJSF1, LJSF2, LJSF3, LJSF4. The sulfate content and total sugar of each fraction were determined according to our published work [25]. LJSF4 was collected for further assays in this work.

#### 3.2.2. Determination of Monosaccharide Composition

Monosaccharide composition of LJSF4 was analyzed by a 6890 N gas chromatographic system (Agilent 6890 N, Agilent, Santa Clara, CA, USA) after hydrolysis and acetylation according to the method described by Zha et al. [43]. 

### 3.3. Effect of LJSF4 on UVB-Irradiated HaCaT Cells

#### 3.3.1. HaCaT Cells Culture and UVB Irradiation

HaCaT cell line was obtained from Korean Cell Line Bank (Seoul, Korea). Cells were cultured in DMEM supplemented with 100 μg/mL of streptomycin, 100 unit/mL of penicillin, and 10% FBS at 37 °C in humidified atmosphere with 5% CO_2_.

Cells were exposed to UVB irradiation at wavelength of 280–320 nm by a UVB meter (UV Lamp, VL-6LM, Vilber Lourmat, Paris, France). Cells were irradiated by UVB at a dose of 30 mJ/cm^2^ [40,42] and incubated until analysis.

#### 3.3.2. Cell Viability Assay

The cytotoxicity of LJSF4 on HaCaT cells were determined by MTT assay. In brief, HaCaT cells were seeded in a 96-well plate at a density of 3 × 10^4^ cells per well. After 24 h, the supernatant was discarded and wells were treated with 25, 50, 100 μg/mL of LJSF4, respectively. Then, after being further incubated for 24 h, the MTT solution was applied into the wells for 4 h. At last, the formazan crystals were dissolved in DMSO, and the absorbance were measured at 540 nm by an ELISA plate reader (BioTek PowerWave XS, Winooski, VT, USA) [10].

#### 3.3.3. Measurement of Intracellular ROS Generation and Cell Viability in UVB-Irradiated HaCaT Cells

Intracellular ROS generation of LJSF4 on UVB-irradiated HaCaT cells was determined by the DCFH-DA method [44]. Briefly, after incubation for 24 h, cells were treated with LJSF4 for 30 min. Then, 500 µg/mL of DCFH-DA was added into each well and incubated for 30 min. Subsequently, cells were exposed to 30 mJ/cm^2^ of UVB and measured according to the reported procedure of Wang et al. [42].

In order to measure the cell viability of LJSF4 on UVB-induced HaCaT cells, cells were treated with LJSF4 (25, 50, 100 μg/mL). After being cultured for 2 h, cells were exposed to 30 mJ/cm^2^ of UVB and incubated for 24 h. Subsequently, cell viability was examined by the MTT assay described previously [10,45].

#### 3.3.4. Measurement of Apoptosis in UVB-Irradiated HaCaT Cells

As in the procedure described by Naito et al., the apoptosis body formation was assessed by nuclear staining [46]. After being seeded in 24-well plates for 24 h, cells were treated with LJSF4 and incubated for another 2 h. Then, cells were exposed to 30 mJ/cm^2^ of UVB and incubated with serum-free DMEM solution for 6 h. After incubation, cells were treated with Hoechst 33342 (stock, 10 mg/mL) for 10 min. At last, the stained cells were photographed by a fluorescence microscope equipped with a Cool SNAP-Procolor digital camera (Olympus, Tokyo, Japan). Apoptosis levels were measured using Image J software automatically.

#### 3.3.5. Western Blot Analysis

To investigate the effect of LJSF4 on the expression of apoptosis-related proteins, Western blot analysis was performed by the method described by Wijesinghe et al. [47]. Briefly, cells exposing to 30 mJ/cm^2^ of UVB were pretreated with different concentrations of LJSF4. After incubation for 24 h, cells were harvested in lysis buffer. The total protein levels of the supernatant were measured using a BCATM kit. A gradient 10% SDS-PAGE was used to separate the total protein extracts (50 μg), which were then electrophoretically transferred onto nitrocellulose membranes. The membranes were then blocked with skim milk (5%), and the immunoblotting procedure, including incubation with primary antibodies, washing, and incubation with secondary antibodies, was carried out as previously reported [47]. Finally, the target bands were observed by enhanced chemiluminescence (ELC), and detected by Tanon-5200 detection system (Tanon Science, Shanghai, China).

### 3.4. Effect of LJSF4 on UVB-Irradiated Zebrafish

#### 3.4.1. Origin and Maintenance of Parental Zebrafish

This study work was approved by the ethical committee of experimental animal care of Ocean University of China at College of Food Science and Engineering, (Approval No. 2019-05). Adult zebrafish were obtained from the Molecular Medicine Laboratory (Ocean University of China, Qingdao, China) and kept in an automatic circulation culture system (ESEN, Beijing, China) at 28.5 ± 1 °C with a cycle of 14/10 h of light/dark. Live brine shrimps were fed twice a day to the parental zebrafish. Embryos were obtained from natural spawning according to the method reported by Zou et al. [48]. Fertilized embryos were collected in petri dishes completely within 30 min.

#### 3.4.2. Measurement of Effect of LJSF4 Against UVB-Irradiation in Zebrafish

At 2 dpf, the zebrafish were transferred to a 24-well plate and treated with LJSF4 with final concentrations of 25, 50, and 100 µg/mL, respectively. The incubation, UVB-irradiation process and determination of ROS level, cell death, and NO production were carried out according to the reported procedure of Ko et al. [12]. The zebrafish larvae were photographed under the microscope Cool SNAP-Procolor digital camera, and the individual zebrafish larvae fluorescence intensity was quantified using an Image J program (National Institutes of Health, Bethesda, MD, US).

### 3.5. Statistical Analysis

Three independent experiments were carried out for each assay. Data were expressed as means ± standard error of the means (S.E.M) and analyzed by one-way ANOVA followed by the least significant differences test using SPSS 20.0 (SPSS Inc., Chicago, IL, USA). Turkey’s multiple-range test was used to assess the significant differences of the means. In all cases, the *p*-value (*p* < 0.05) was indicated as statistically significant differences.

## 4. Conclusions

In this study, we evaluated the protective effect of LJSF4 against UVB-induced photodamage by in vitro and in vivo models. Treatment with LJSF4 prior to UVB exposure considerably protected the HaCaT cells against ROS generation. In addition, the ROS, the cell death, the nitric oxide, and the lipid peroxidation induced by UVB radiation in zebrafish were reduced by the addition of LJSF4. This study suggested the potential use of LJSF4 from *S. japonica* for the treatment of UVB-caused skin damage, which indicated that LJSF4 has a great potential to be used in cosmetic products as an effective natural anti-UVB agent.

## Figures and Tables

**Figure 1 marinedrugs-18-00316-f001:**
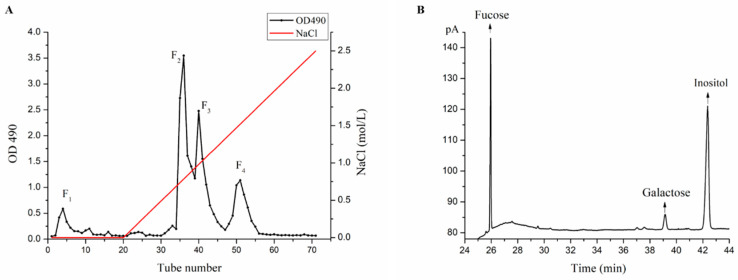
Purification and monosaccharide determination of fucoidan fractions: (**A**) elution profile of *S. japonica* crude fucoidan on DEAE-Sepharose Fast Flow anion exchange chromatography; (**B**) monosaccharide compositions of LJSF4 determined by gas chromatography.

**Figure 2 marinedrugs-18-00316-f002:**
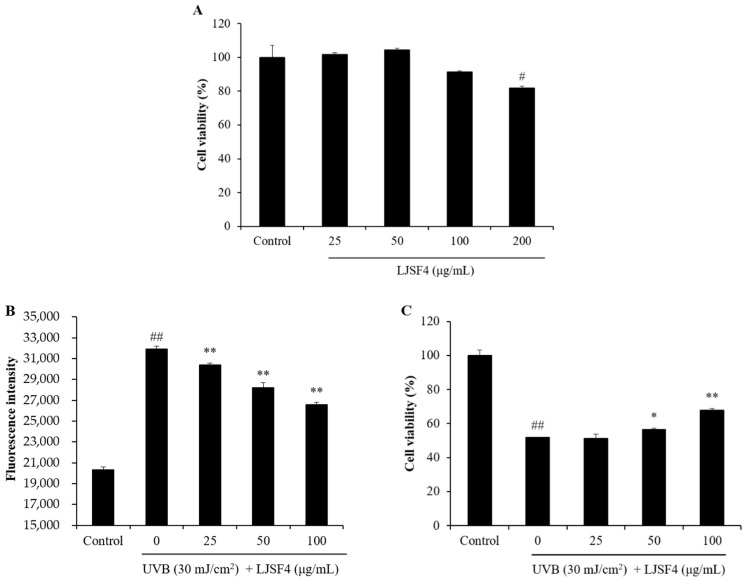
Protective effect of LJSF4 on UVB-induced HaCaT cells: (**A**) cytotoxicity of LJSF4 on HaCaT cells; (**B**) intracellular ROS level of UVB-irradiated HaCaT cells; (**C**) the viability of UVB-irradiated HaCaT cells. All experiments were performed in triplicate. Data are expressed as means ± standard error of the means (S.E.M). * *p* < 0.05, ** *p* < 0.01 as compared to the UVB-treated group and ^#^
*p* < 0.05, ^##^
*p* < 0.01 as compared to the control group.

**Figure 3 marinedrugs-18-00316-f003:**
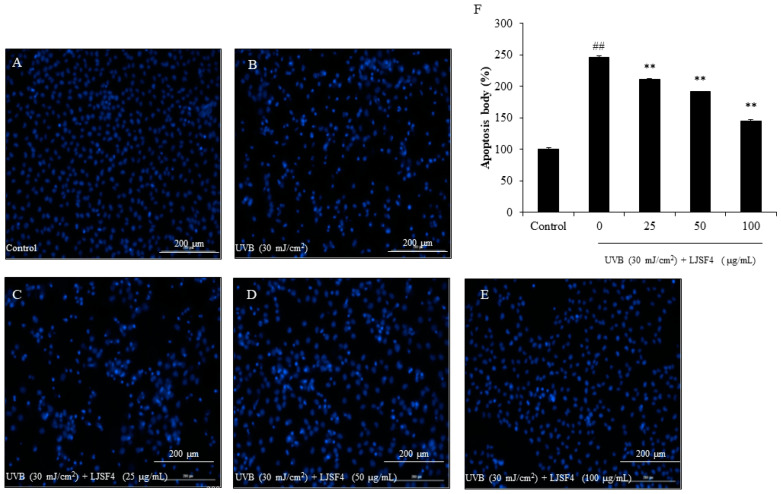
The apoptotic body formation levels in UVB-irradiated HaCaT cells: (**A**) nuclear morphology of non UVB-irradiated HaCaT cells; (**B**) nuclear morphology of UVB-irradiated HaCaT cells; (**C**) nuclear morphology of cells treated with 25 of µg/mL LJSF4 and irradiated with UVB; (**D**) nuclear morphology of cells treated with 50 of µg/mL LJSF4 and irradiated with UVB; (**E**) nuclear morphology of cells treated with 100 of µg/mL LJSF4 and irradiated with UVB; (**F**) reactive apoptotic body formation. Apoptosis levels were measured using Image J software. All experiments were performed in triplicate. Data are expressed as means ± standard error of the means (S.E.M). ** *p* < 0.01 as compared to the UVB-treated group and ^##^
*p* < 0.01 as compared to the control group.

**Figure 4 marinedrugs-18-00316-f004:**
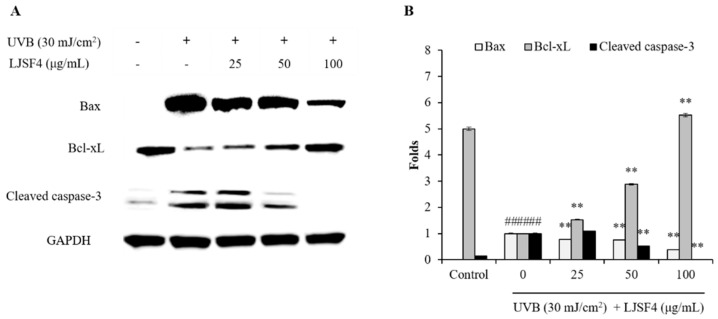
Effect of LJSF4 on Bax/Bcl-xL, and cleaved caspase-3 levels in UVB-irradiated HaCaT cells: (**A**) the effect of LJSF4 on UVB-induced apoptosis related protein expression; (**B**) the relative amounts of Bax/Bcl-xL and cleaved caspase-3 levels compared with GAPDH. The Bax level of control group and the cleaved caspase-3 level of experimental group treated with 100μg/mL of LJSF4 were not detected. All experiments were performed in triplicate. Data are expressed as means ± standard error of the means (S.E.M). ** *p* < 0.01 as compared to the UVB-treated group and ^##^
*p* < 0.01 as compared to the control group.

**Figure 5 marinedrugs-18-00316-f005:**
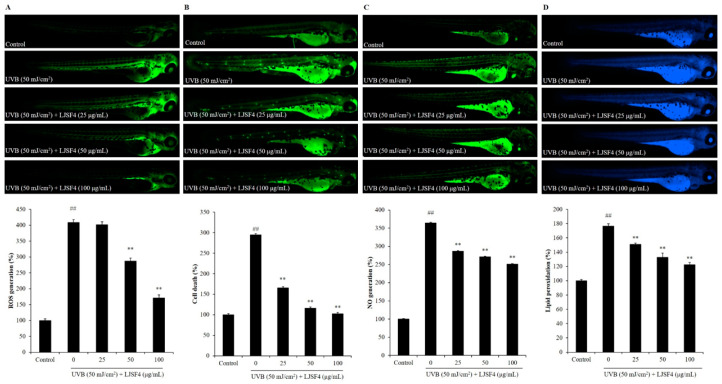
The effect of LJSF4 on UVB-irradiated zebrafish: (**A**) effect of LJSF4 on UVB-irradiated ROS generation in zebrafish; (**B**) effect of LJSF4 on UVB-irradiated cell death levels in zebrafish; (**C**) effect of LJSF4 on UVB-irradiated NO level in zebrafish; (**D**) effect of LJSF4 on UVB-irradiated lipid peroxidation levels in zebrafish. Zebrafish embryos at 2 days post-fertilization (dpf) were used for the anti-UVB study. Data are expressed as means ± standard error of the means (S.E.M). ** *p* < 0.01 as compared to the UVB-treated group and ^##^
*p* < 0.01 as compared to the control group.

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
