# Peer review of "Protective Effect of a Fucose-Rich Fucoidan Isolated from Saccharina japonica against Ultraviolet B-Induced Photodamage In Vitro in Human Keratinocytes and In Vivo in Zebrafish"

_marinedrugs, 2020, doi:10.3390/md18060316_

Round 1

Reviewer 1 Report

The manuscript deals with interesting topic and is well written. Obtained results are novel, important and applicable.

However, there are present some points which need to be corrected:

  1. Fig. 1 is not of enough quality and letters are too small to read them.
  2. Fig. 3: the size bar is not readable. There will be better to mention the size of the bar in figure caption here.
  3. Lines 211, 221, 226, 242: According to the guide for authors, for US suppliers, there should be given city and state of origin as well. (as it is mentioned on line 210, for example).
  4. lines 259 as well as 288: city of origin for digital camera is missing. Moreover, you have to unify the name of the instrument (Pro color vs. Procolor). 

All in all, only minor issues have to be corrected and manucript will be ready to be accepted.

Author Response

Thanks for your keen revision. The manuscript has been revised according to your comments and high-light in red.

Point 1: Fig. 1 is not of enough quality and letters are too small to read them.

Response 1: The figure 1 was adjusted to enough quality and letters were enlarged.

Point 2: Fig. 3: the size bar is not readable. There will be better to mention the size of the bar in figure caption here.

Response 2: The figure 3 was modified by enlarging the size bar which is 200 μm.

Point 3: Lines 211, 221, 226, 242: According to the guide for authors, for US suppliers, there should be given city and state of origin as well. (as it is mentioned on line 210, for example).

Response 3: Thanks a lot! All the city and state of origin for suppliers were added according to your comments. (Line 216, 220, 229, 234, 242, 250)

Point 4: Lines 259 as well as 288: city of origin for digital camera is missing. Moreover, you have to unify the name of the instrument (Pro color vs. Procolor).

Response 4: Thanks for your reminder! The city of origin for digital camera was added, and the name of the instrument was unified as Procolor. (Line 268, 295)

Reviewer 2 Report

This is an interesting research which worth publication. The manuscript can be further improved by addressing the following points:

Title: Protective Effect of a Fucose-Rich Fucoidan Isolated from Saccharina japonica against Ultraviolet B-Induced Photodamage In Vitro in Human Keratinocytes and In Vivo in Zebrafish

1. The words "in vitro" and "in vivo" need italic letters

2. Line 56: "Saccharina japonica (S. japonica), formerly known as Laminaria japonica (L. japonica), is an edible brown seaweed, which is plentifully produced in Asian countries and utilized as a traditional Chinese medicine and functional food [14]" a reference to support that Saccharina japonica and Laminaria japonica is the same seaweed is necessary.

3. Line 244, the passage of cells should be write in the text.

4. Line 250, the tested concentration of LJSF4 was 25, 50 and 100 mM, but in the results section the concentration was in ug/mL. What is correct?

Author Response

Thanks for your keen revision. The manuscript has been revised according to your comments and high-light in blue.

Point 1: The words "in vitro" and "in vivo" need italic letters.

Response 1: Thanks a lot! All of the words "in vitro" and "in vivo" were revised according to your comments. (Line 4, 5)

Point 2: Line 56: "Saccharina japonica (S. japonica), formerly known as Laminaria japonica (L. japonica), is an edible brown seaweed, which is plentifully produced in Asian countries and utilized as a traditional Chinese medicine and functional food [14]" a reference to support that Saccharina japonica and Laminaria japonica is the same seaweed is necessary.

Response 2: A reference to support that Saccharina japonica and Laminaria japonica is the same seaweed was added. (Line 58)

Reference:

Liu, F.; Wang, X.; Yao, J.; Fu, W.; Duan, D., Development of expressed sequence tag-derived microsatellite markers for Saccharina (Laminaria) japonica. J. Appl. Phycol. 2009, 22, (1), 109-111.

Point 3: Line 244, the passage of cells should be written in the text.

Response 3: The passage of cells was written in line 238.

Point 4: Line 250, the tested concentration of LJSF4 was 25, 50 and 100 mM, but in the results section the concentration was in µg/mL. What is correct?

Response 4: The typing error was revised according to your comments. All units were unified into “µg/mL”. (Line 247, 259)

Reviewer 3 Report

This paper deals with the protective Effect of a Fucose-Rich Fucoidan Isolated from Saccharina japonica against Ultraviolet B4 Induced Photodamage In Vitro in Human Keratinocytes and In Vivo in Zebrafish. The topic is interesting and falls within the Journal scope. The current version of the manuscript is well presented and organized, however some lacks should be improved. Please, follow the comments below:

Keywords: Personal note: Words included in the title should be avoided in this section, in order to expand the visibility of the manuscript.

Abstract: Acronyms should be avoided.

Materials and methods: Replicates should be included in all sections.

Further justification of the used conditions or the corresponding reference should be included in this section.

Discussion should be extended. Further details and comparison with recent studies in the literature are missed.

Author Response

Thanks for your keen revision. The manuscript has been revised according to your comments and high-light in purple.

Point 1: Keywords: Personal note: Words included in the title should be avoided in this section, in order to expand the visibility of the manuscript.

Response 1: The keywords were changed to be “ROS; Phaeophyta; carbohydrate; UVB irradiation; HaCaT cells; zebrafish”, in order to expand the visibility of the manuscript. (Line 31)

Point 2: Abstract: Acronyms should be avoided.

Response 2: Thanks a lot! The acronyms of abstract were revised according to your comments.

“LJSF4” was changed to “the purified fucoidan”.

“ROS” was changed to “re­active oxygen species”.

Point 3: Materials and methods: Replicates should be included in all sections.

Response 3: All experiments were carried out in triplicate, and per experiment had at least three parallel groups. Replicates were shown in “3.5. Statistical Analysis” (Line 298) and all the related figure legends.

Point 4: Further justification of the used conditions or the corresponding reference should be included in this section.

Response 4: All the used conditions and corresponding references were common methods which were suitable for our research.

Point 5: Discussion should be extended. Further details and comparison with recent studies in the literature are missed.

Response 5: Thanks for your suggestion. Discussion of further details and comparison with recent studies were added (line 126 to 131, line 192 to 199).

Reviewer 4 Report

The study by Su, et al. reports on a protective effect of a fucose-rich fucoidan isolated from the seaweed S. japonica against UVB light. This study build on a recently published paper from the same group where they addressed the structure and anti-inflammatory properties of this compound. Using a combination of in vitro assays with a keratinocyte cell line and in vivo assays with zebrafish, they demonstrate that this compound can block UVB-induced ROS production and cell death. In line with this, they show that the fucoidan blocks UVB-induced caspase-3 cleavage. The UVB protective effect of this compound is nicely demonstrated, and in line with previous findings with similar compounds from other species. I support publication, and only have some minor comments that the authors should consider.

1-My main comment is that I would like to have some proposed mechanism of how LJSF4 protects against UVB. Does the LJSF4 directly absorb UVB light, thus act as a sunscreen? Or does it act as an anti-oxidant, for instance by scavenging free electrons? The authors should add a brief discussion section explaining/speculating on the mechanism. It will also be worthwhile to explain how this compound can be applied in the cosmetic industry. Topical administration as a sunscreen?

2-Related to previous comment: does LJSF4 absorb UVB-light? If so, can the authors provide an absorption spectrum?

3-Lines 89-90: "According to the results, cell proliferation was not significantly affected by ...". MTT is not a readout of cell proliferation, but measures activity of NADH-dependent cellular oxidoreductases. This is only an indirect readout of cell viability, and definitely not of cell proliferation. The text needs to be corrected at this point.

4-Which ROS assay was used in Figure 2B? Explain which ROS probe was used in the main text or figure legend.

5-The microscopy images of figure 3A-E are very small. In addition to the overview images, magnifications of representative cells with/without apoptotic body formation need to be shown such that the differences can be appreciated.

6-The authors write that apoptosis levels were measured using ImageJ software (line 134). Particularly how was this done? Was this analysis manual or automated? Please provide details of the analysis in the Methods section 3.3.4.

7-Throughout the manuscript, the authors write caspass-3. I assume they mean caspase-3 (with an 'e')?

8-In the methods section (lines 211-213), provide details of all antibodies used (clone and catalog numbers).

Author Response

Thanks for your keen revision. The manuscript has been revised according to your comments and high-light in green.

Point 1: My main comment is that I would like to have some proposed mechanism of how LJSF4 protects against UVB. Does the LJSF4 directly absorb UVB light, thus act as a sunscreen? Or does it act as an anti-oxidant, for instance by scavenging free electrons? The authors should add a brief discussion section explaining/speculating on the mechanism. It will also be worthwhile to explain how this compound can be applied in the cosmetic industry. Topical administration as a sunscreen?

Response 1: Thanks for your comments. In our opinions, LJSF4 does not absorb UVB directly, but act as an antioxidant. As we described in the methodology, HaCaT cells were treated with LJSF4 for a short time and the LJSF4-treated HaCaT cells were washed with PBS before exposure to UVB. LJSF4 was absorbed by cells or bonded to a membrane receptor and transfer the signal into the cell, then stimulate the intracellular defense system. Non-absorbed or non-bonded sample was removed by PBS washing before UVB irradiation. Besides, the results indicate that LJSF4 suppresses UVB-induced cellular damage through regulation of the expressions of apoptotic-related proteins by scavenging intracellular ROS. Thus, we thought the mechanism of UV protective effect of LJSF4 was though suppressing oxidative stress.

Point 2: Related to previous comment: does LJSF4 absorb UVB-light? If so, can the authors provide an absorption spectrum?

Response 2: As we replay above, LJSF4 does not absorb UVB, but act as an antioxidant.

Point 3: Lines 89-90: "According to the results, cell proliferation was not significantly affected by ...". MTT is not a readout of cell proliferation, but measures activity of NADH-dependent cellular oxidoreductases. This is only an indirect readout of cell viability, and definitely not of cell proliferation. The text needs to be corrected at this point.

Response 3: Thanks for your suggestion. The discussion of the results was corrected and the words of cell proliferation were deleted.

Point 4: Which ROS assay was used in Figure 2B? Explain which ROS probe was used in the main text or figure legend.

Response 4: According to method 3.3.3., intracellular ROS generation of LJSF4 on UVB-irradiated HaCaT cells was determined by DCFH-DA method. The main text used DCFH-DA ROS probe. The description was added in Line 253 to 257.

Point 5: The microscopy images of figure 3A-E are very small. In addition to the overview images, magnifications of representative cells with/without apoptotic body formation need to be shown such that the differences can be appreciated.

Response 5: The microscopy images of figure 3A-E were modified according to your comments.

Point 6: The authors write that apoptosis levels were measured using Image J software (line 134). Particularly how was this done? Was this analysis manual or automated? Please provide details of the analysis in the Methods section 3.3.4.

Response 6: Image J software can be used in determining the apoptosis levels. It is an automated analysis. The description was added in section 3.3.4. (Line 268 to 269)

Point 7: Throughout the manuscript, the authors write caspass-3. I assume they mean caspase-3 (with an 'e')?

Response 7: The typing error was revised according to your comments. All words were unified to “caspase-3”. (Line 141 to 165)

Point 8: In the methods section (lines 211-213), provide details of all antibodies used (clone and catalog numbers).

Response 8: The details of all antibodies used (clone and catalog numbers) were added. (line 216 to 219)

Round 2

Reviewer 2 Report

Now, the manuscript has been improved.